# Creation and annihilation of topological meron pairs in in-plane magnetized films

N. Gao[1,2], S.-G. Je[3], M.-Y. Im[3,4], J.W. Choi[5], M. Yang[2], Q. Li[2], T.Y. Wang[2], S. Lee[6], H.-S. Han[6], K.-S. Lee [6], W. Chao[3], C. Hwang[7], J. Li[8]* & Z.Q. Qiu [2]*

Merons which are topologically equivalent to one-half of skyrmions can exist only in pairs or groups in two-dimensional (2D) ferromagnetic (FM) systems. The recent discovery of meron lattice in chiral magnet $Co_8Zn_9Mn_3$ raises the immediate challenging question that whether a single meron pair, which is the most fundamental topological structure in any 2D meron systems, can be created and stabilized in a continuous FM film? Utilizing winding number conservation, we develop a new method to create and stabilize a single pair of merons in a continuous Py film by local vortex imprinting from a Co disk. By observing the created meron pair directly within a magnetic field, we determine its topological structure unambiguously and explore the topological effect in its creation and annihilation processes. Our work opens a pathway towards developing and controlling topological structures in general magnetic systems without the restriction of perpendicular anisotropy and Dzyaloshinskii–Moriya interaction.

[1] Key Laboratory of Microelectronic Devices and Integrated Technology, Institute of Microelectronics of Chinese Academy of Sciences, Beijing 100029, China.
[2] Department of Physics, University of California at Berkeley, Berkeley, CA 94720, USA. [3] Center for X-ray Optics, Lawrence Berkeley National Laboratory, Berkeley, CA 94720, USA. [4] Department of Emerging Materials Science, DGIST, Daegu, Korea. [5] Center for Spintronics, Korea Institute of Science and Technology, Seoul 02792, Korea. [6] School of Materials Science and Engineering, Ulsan National Institute of Science and Technology, Ulsan 44919, Korea. [7] Korea Research Institute of Standards and Science, Yuseong, Daejeon 305-340, Korea. [8] International Center for Quantum Materials, School of Physics, Peking University, Beijing 100871, China. *email: jiali83@pku.edu.cn; qiu@berkeley.edu

Topology has been recognized to play an important role in condensed matter physics since the theoretical discoveries of topological phase transitions and topological phases of matter[1–3]. While manifesting usually in reciprocal space, where the nontrivial wrapping of the Brillion zone around the Hamiltonian space leads to exotic topological states[4], topological structures also exist in real space as localized spin textures of a two-dimensional (2D) Heisenberg system with a topological number of

$$N = \frac{1}{4\pi} \int \mathbf{n} \cdot \left( \frac{\partial \mathbf{n}}{\partial x} \times \frac{\partial \mathbf{n}}{\partial y} \right) dx dy \qquad (1)$$

where $\mathbf{n}$ is the direction vector of magnetization[5]. In a continuous film, $N$ takes only integer values corresponding to different homotopy classes of the $S^2$ spin space wrapped by the whole 2D plane. Since the discovery of stable skyrmions ($N = \pm 1$)[6–8] in perpendicularly magnetized ferromagnetic (FM) films consisting of the Dzyaloshinskii–Moriya interaction (DMI)[9,10] due to inversion symmetry breaking, intensive studies[11–19] have been carried out over the last decade on the creation, manipulation, and detection of skyrmions toward future spintronic technologies[20].

In addition to the discovery of skyrmions in perpendicularly magnetized systems, there has also been a great effort in searching for new forms of topological structures in in-plane magnetized systems, which are more accessible in experiment due to the intrinsic demagnetization field in all magnetic films[21]. One promising structure is the so-called meron, which was originally described in the context of quark confinement[22] and later identified in condensed matter physics, as a magnetic vortex and topologically equivalent to one-half of a skyrmion. (Note that an important difference between merons and skyrmions is their peripheral spin textures: those of merons align in the in-plane while those of skyrmions point toward out-of-plane directions.) However, an individual meron is not localized and thus can only exist in confined geometries[23–27]. In a continuous film, merons must exist in pairs or groups[28–35]. Experimentally, multiple vortices were observed only as transient states[36,37] or in aggregated groups[27,38–40]. It was only recently that topologically nontrivial meron lattice was observed in chiral magnet $Co_8Zn_9Mn_3$[41] in the form of square lattices.

There are several major obstacles preventing the development of topological structures in in-plane magnetized systems. Firstly, the meron pair, which is the most concise and fundamental localized topological structure in in-plane magnetized FM films, has not yet been observed unambiguously[42]. Second, chiral magnets usually require a precise control of the material compositions and crystal structures[41,42] to tune the easy magnetization axis and DMI strength, further limiting the availability of material systems[13,19,43–46]. Third, topological information is mostly carried by spin textures around the meron core area[35], whereas most of magnetic imaging techniques can hardly provide a spatial resolution to directly image the meron core polarity, especially in the presence of a magnetic field.

In this paper, we report the stabilization of a meron pair in a continuous in-plane magnetized Py film. By a direct magnetic imaging of the out-of-plane core magnetization using the full-field magnetic transmission soft X-ray microscopy (MTXM), we determine its topology unambiguously. Moreover, the creation, evolution, and annihilation of the meron pair are explored in details to reveal the topological effects in the magnetization process.

## Results

**Methodology for meron pair stabilization.** The meron topology (Fig. 1a) is determined by the winding number ($w$), which describes the in-plane magnetization rotational direction ($w = +1$ for vortex, $w = -1$ for antivortex) and the polarity ($p$), which describes the out-of-plane core magnetization ($p = +1$ for up and $p = -1$ for down magnetization of the core)[23], with the topological number being exactly one-half of the product of polarity and winding number ($N = pw/2$)[35]. Following the definition in literature, $N = -1/2$ defines a meron and $N = +1/2$ defines an antimeron[41] (the relation between vortex/antivortex and meron/antimeron is shown in Fig. 1a).

To realize a meron pair in a 2D system, the most challenging step is how to deform the spin texture delicately and reliably to produce exactly two merons? Here, we utilize the fact that the winding number is conserved[1] in in-plane magnetized systems. Thus with a homogeneous background ($w = 0$), a creation of local magnetic vortex ($w = +1$) must be accompanied by an antivortex ($w = -1$) spontaneously to conserve the total winding number of zero. In experiment, such local vortex creation could be achieved by magnetic imprinting[17–19] of a vortex state from a FM disk.

To explore the feasibility of this methodology, a micromagnetic simulation was performed on a system with a Co disk on top of a continuous Py thin film (Fig. 1b). The result shows clearly the stabilization of a vortex state in the Co disk, although the vortex core is shifted away from the disk center (Fig. 1c) due to the coupling of the vortex with surrounding Py magnetization which, to certain extent, is equivalent to applying a magnetic field to the vortex[47]. The more important observation is that the Co disk not only imprints the vortex state into the Py film, but also creates an antivortex simultaneously in the Py film next to the vortex core due to the conservation of the total winding number. We would like to emphasize that the vortex–antivortex pair here is a stable meron pair well localized by the disk area.

The total topological number of the meron pair is

$$N = \frac{1}{2} p_V w_V + \frac{1}{2} p_A w_A = \frac{1}{2}(p_V - p_A) \qquad (2)$$

where $p_v$ ($p_A$) and $w_V$ ($w_A$) denote the polarity and winding number of the vortex (antivortex), respectively. Therefore, the meron pair (Fig. 1d) can be either a bimeron ($N = \pm 1$) for antiparallel alignment or a meron–antimeron pair ($N = 0$) for parallel alignment of the two cores. These two different topological states can be intuitively recognized by a continuous spin deformation: a global $\pi/2$ rotation around $-Sy$-axis followed by another $\pi/2$ rotation around $+Sz$-axis in the spin space. After this two-step spin deformation, a bimeron state with $N = -1$ is transformed into a $N = -1$ skyrmion, in which the full $S^2$ spin space is wrapped by the whole film plane (Fig. 1e). In contrast, a meron–antimeron pair is transformed into a state, in which the $y > 0$ ($y < 0$) half plane wraps (unwraps) the same half of the $S^2$ spin space, allowing a continuous shrinking of this semisphere to a single point on $S^2$, i.e., a single domain state with $N = 0$ (Fig. 1f). In our approach, the Py film can be easily manipulated by magnetic field, providing a practical way to tailor the polarity and thus the topology of the meron pair.

**Experimental observation of topological meron pairs.** Arrays of 40 nm thick and 1 μm radius Co disks with 3 μm center-to-center distance (Fig. 2a) were grown on top of a continuous 80 nm thick Py film (see Methods). In the region without the Co disks, the Py film exhibits a typical soft magnetic hysteresis loop with a small coercivity $Hc$ (~20 Oe) and ~100% remanence (Fig. 2b). In the patterned area, where there is a mixture of Py film and Co disks, the hysteresis loop remanence is reduced to ~90% with the

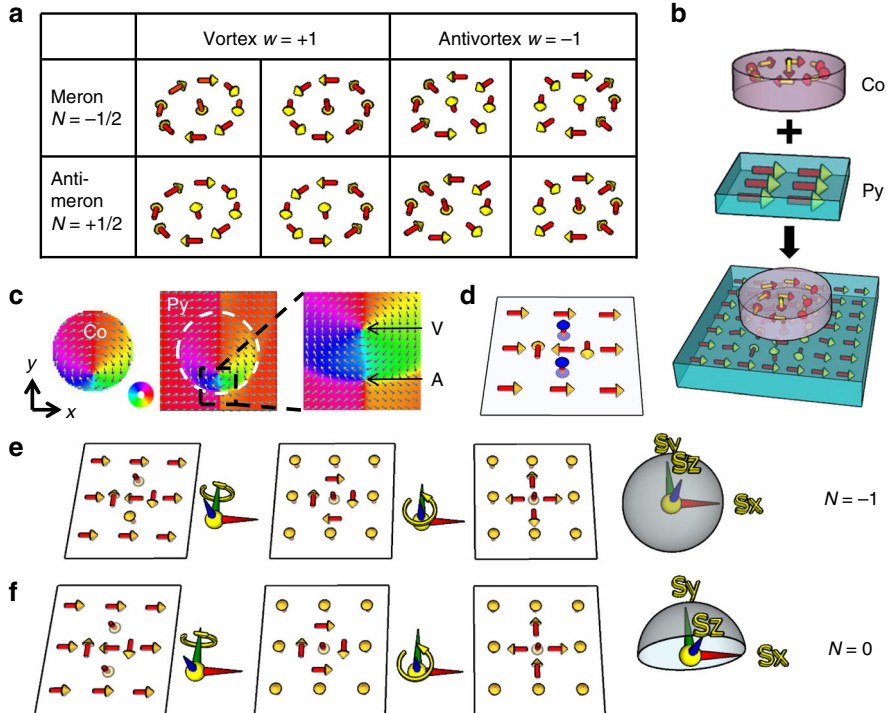

**Fig. 1 Meron pair and topology. a** Definition and corresponding relation between vortice/antivortice and meron/antimeron, where $w$ denotes the winding number and $N$ denotes the topological number ($N = pw/2$ with $p = +1$ for up and $p = -1$ for down polarity of the cores). It is worth to mention that winding number is always +1 for vortex and −1 for antivortex, and does not depend on the clockwise or counterclockwise circulation direction of the in-plane magnetization[35]. **b** Schematic drawing of vortex imprinting from a Co disk into a continuous Py film. **c** Simulation result of the magnetization distribution in a system with a Co disk (40 nm thick and 1 μm radius) on top of a 80 nm thick Py film. The result clearly shows the imprinting of the vortex from Co into Py, as well as a spontaneous creation of an antivortex to pair with the imprinted vortex. The positions of the vortex and antivortex in the Py film are highlighted by "V" and "A" in the zoomed in magnetization profile. The color wheel is shown next to the Co profile (each color on the wheel represents magnetization directing toward the corresponding radial direction outward.) **d** A simplified illustration of the magnetization profile of the meron pair. **e, f** Topology of the meron pair depends on their core polarities. Under a continuous spin deformation, the bimeron state (**e**; $N = -1$) is transformed into a skyrmion which wraps the full $S^2$ spin space. In contrast, the meron–antimeron pair (**f**; $N = 0$) is transformed into a state in which the $y > 0$ ($y < 0$) half plane wraps (unwraps) the same half of the $S^2$ spin space, allowing a continuous shrinking of this semisphere to a single point on $S^2$, i.e., a single domain state with $N = 0$. Note, there are totally four possible combinations of the core polarities. We only show two of them because the other two cases are just mirror reflection of the shown two cases.

saturation field $H$s increased to above 150 Oe, indicating the effect of the Co disks.

To experimentally confirm the methodology of the vortex imprinting, one needs to measure the magnetization profiles of the Co disk and the Py film separately. For this purpose, we used MTXM that enables element-resolved magnetic imaging (e.g., imaging Co and Py spin structures separately), by tuning the X-ray photon energy to the 2p core-level absorption energy of the corresponding element[48]. The MTXM spatial resolution could reach 25 nm by the state-of-the-art X-ray optics of the so-called Fresnel zone plates[49] (Supplementary Fig. 1). The magnetic contrast comes from the X-ray magnetic circular dichroism (XMCD) mechanism[50]: the absorption of a circularly polarized X-ray by a FM sample depends on the angle between the magnetization direction of the sample and the photon spin direction of the X-ray. At a fixed incident angle of a circular polarized X-ray beam, different magnetization directions of a magnetic domain then give rise to different contrasts (dark/grey/bright) of the MTXM image.

We first mounted the sample with the in-plane geometry to verified vortex imprinting. In this geometry, the area having the magnetization pointing to $+x$ ($-x$) gives black (bright) contrast, while that pointing toward $+y$ ($-y$) direction shows grey contrast (Supplementary Fig. 2). Figure 2c, f shows the representative

Co and Py domain images taken at the Co (778.0 eV) and Fe (707.5 eV) absorption edges, respectively. As shown in Fig. 2c, the upper (lower) part of the Co disk exhibits a dark (bright) contrast, while the contrast on the left and right parts of the disk changes gradually in between, which are more clearly visible in the corresponding simulated images (Fig. 2d, e). Such a configuration corresponds to a typical vortex structure, where the in-plane magnetization forms a closed loop, around a tiny vortex core. Here the core position is shifted off the center of the disk, in agreement with the simulation result (Fig. 1c). More importantly, the Py beneath the Co disk also exhibits an almost identical magnetic contrast inside the disk region, indicating a vortex state with identical core position (Fig. 2f–h), and proving that the Co/Py magnetic coupling has imprinted the vortex state from the Co disk into the Py continuous film.

The antivortex accompanied with the vortex is expected to be confined to a very small region in the Py film outside the disk area (Fig. 1c), making it difficult to be observed in in-plane magnetic images (Fig. 2f). Noticing that the out-of-plane core magnetization of a vortex or an antivortex has the greatest magnetic contrast relative to the surrounding magnetization, and that topology can only be determined by information of the core polarity [Eq. (2)], we imaged the vortex and antivortex cores by taking out-of-plane magnetic contrast (z-component;

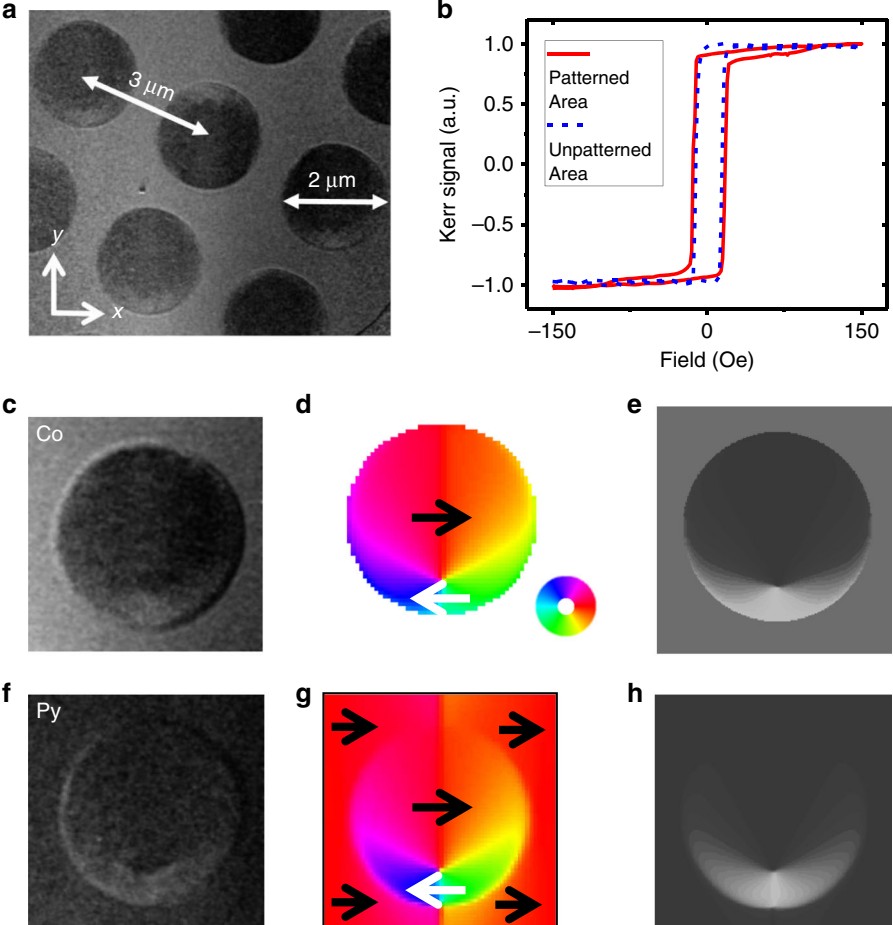

**Fig. 2 Sample structure and vortex imprinting. a** Sample geometry imaged by MTXM at Co L3 edge. **b** Hysteresis loops of the sample inside and outside the patterned area. **c** The in-plane magnetic contrast at Co edge at external field of 50 Oe. **d** The corresponding simulated magnetization profile in Co layer, with the color wheel shown next to it. **e** The $x$-component (horizontal) of magnetization in Co layer from simulation. **f** The in-plane magnetic contrast at Fe edge at external field of 50 Oe. **g** The corresponding simulated magnetization profile in Py layer. **h** The $x$-component (horizontal) of magnetization in Py layer from simulation.

Supplementary Fig. 2). In this mounting geometry, dark (bright) contrast corresponds to $z$-component of magnetization orienting up (down) and grey contrast indicates in-plane magnetization. To achieve all possible states, we repeated the cycles of saturating the sample with an external field of $H = -430$ Oe followed by a flipping of the surrounding Py magnetization, with a field of $H = +35$ Oe. The $H = -430$ Oe field saturates both the Py disk area and the surrounding area magnetizations to the $-x$ direction, while the subsequent $H = +35$ Oe field is greater than the Py coercivity to switch the surrounding Py magnetization from $-$ to $+x$ direction, but less than the saturation field of the vortex. We expected such a process would produce sufficient randomness to get all possible types of meron pairs. Figure 3a–hdepicts the obtained images. As shown in the images, after each cycle, most parts of the Py film shows grey contrast, corresponding to in-plane magnetization. On top of this in-plane background, we always observed two small dots (dark or bright; ~100 nm diameter) close to either the upper or lower edge of the disk region. These dots correspond to the meron cores, which have an out-of-plane magnetization. The dot inside the disk region corresponds to the vortex core, and the dot just outside the disk region is the antivortex core. The two cores are always bonded together (~300 nm distance) and located near the edge of the disk to have the majority of the vortex in-plane

magnetization being parallel to the surrounding in-plane magnetization, as predicted above in Fig. 1c. The polarities of the vortex and antivortex cores appear randomly, i.e., all four possible up/down combinations of the two core polarities were observed after each cycle with almost equal probabilities. According to Eq. (2), we identified the observed four configurations to either bimerons for antiparallel core polarities ($N = \pm 1$; Fig. 3a, b) or meron–antimeron pairs for parallel core polarities ($N = 0$; Fig. 3c, d). Therefore, we conclude that meron pairs of different topologies are stabilized in the continuous Py film.

**Imaging the magnetization process of the meron pairs.** After creating and identifying stable meron pairs, we investigate their magnetization process by direct magnetic imaging as a function of magnetic field in three different regions: (1) from $H = -Hs$ to $H < +Hc$; (2) $H \sim Hc$; and (3) from $H > +Hc$ to $H = +Hs$. We first saturated the sample with a negative magnetic field, and then took Py in-plane magnetization images in a sequence of $H = 8$ Oe, 40 Oe, and 300 Oe to represent the typical end states of these three regions, respectively (Fig. 4a, b, c). We also took the Py out-of-plane magnetization images in a similar but finer sequence after saturation to resolve the meron cores (Fig. 4j). At 8 Oe ($H < +Hc$; Fig. 4a), the Py in-plane magnetization surrounding the disk region remains in its originally saturated direction (to the

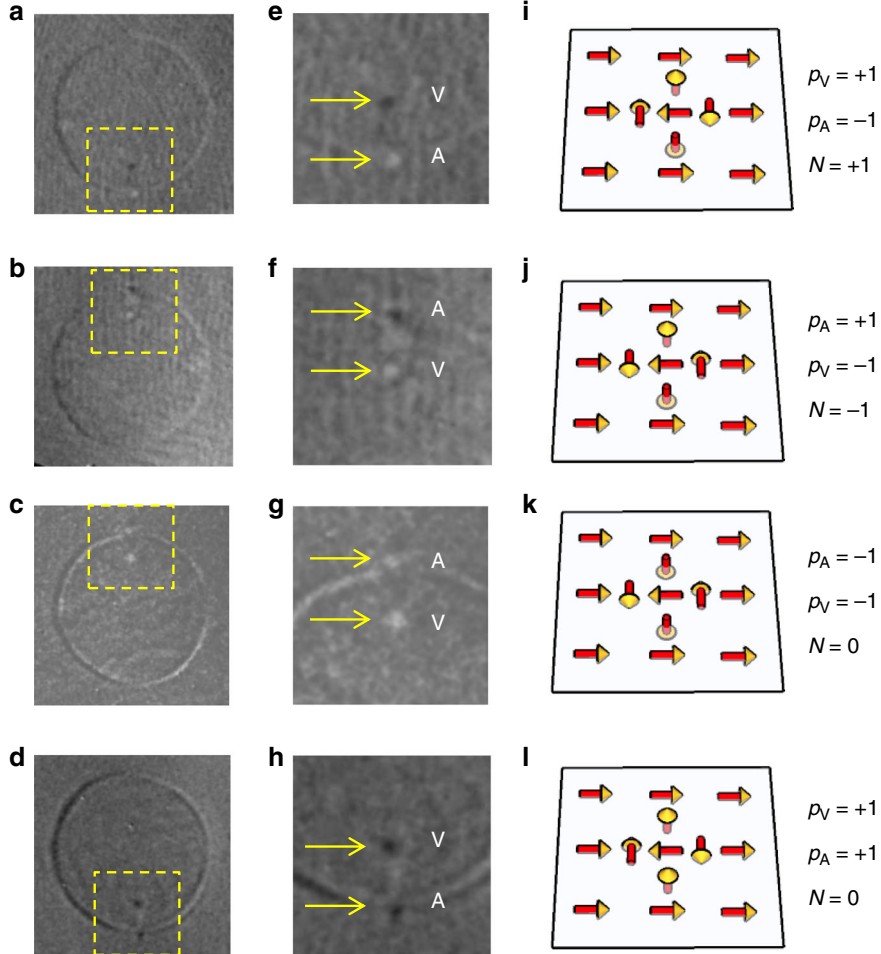

**Fig. 3 Direct observation of the out-of-plane cores of the meron pairs. a–d** Typical out-of-plane magnetic contrast images at 35 Oe. **e–h** Corresponding zoomed in images around the cores of the meron pairs. "V" and "A" denote the vortex core and antivortex core, respectively. **i–l** Corresponding schematic views of the magnetization profiles of the meron pairs. All four combinations of the core polarities are observed. The two cores are always bonded together and locate near either the top or bottom edge of the disk (the choice depends on the in-plane circulation direction of the vortex).

left direction of the image). Inside the disk area, however, the originally single domain state has developed into a vortex state with the core position shifted toward the lower edge of the disk area, as a result of counterclockwise circulation of the in-plane magnetization. As confirmed by a corresponding out-of-plane image (the 15 Oe image of Fig. 4j), there also exists an antivortex core located outside the disk area near the vortex core to form a meron pair. This corresponds to the end state of region (1). At 40 Oe ($+Hc < H < +Hs$; Fig. 4b), the surrounding in-plane Py magnetization has switched to the right direction of the image, resulting in a relocation of the original vortex core from the lower to the top edge of the disk in the image, to match the new direction of the surrounding Py magnetization. During this process, the original antivortex near the lower edge of the disk area has been annihilated accompanied by a simultaneous creation of a new antivortex near the top edge of the disk area, as confirmed by a corresponding out-of-plane image (the 45 Oe image of Fig. 4j). This corresponds to the end state of region (2). As discussed above in Fig. 3, the newly formed meron pair at this state has equal probability to be a bimeron with $N = \pm 1$, or a meron–antimeron pair with $N = 0$ (here for Fig. 4j it chooses to be a bimeron with $N = \pm 1$). In region (3), the distance between the two cores gradually reduces from ~300 nm toward zero with increasing the applied field (the 45–155 Oe images in Fig. 4j), and

the meron pair is finally annihilated at the end of region (3) (Fig. 4c and the 175 Oe image of Fig. 4j). The elongated deformation of the meron cores with increasing field (particularly 85–155 Oe images of Fig. 4j) is a characteristic feature observed recently in Py[51]. As shown in Supplementary Note 2 and Supplementary Figure 3, this does not change their topologies which are robust against continuous deformations.

**The topological effects in the magnetization process.** We explored the topological effect in the above mentioned three different regions of the magnetization process.

In the region of $H = -Hs$ to $H < +Hc$, the Py magnetization surrounding the disk region remains unchanged and the disk region (including nearby region) changes from a uniform magnetization to a meron pair near the edge of the disk boundary as the magnetic field changes gradually. We find that the core polarities of the created meron pair are always parallel to each other (e.g., the 15 Oe image of Fig. 4j), either along $+z$ or $-z$ directions. Recalling that parallel polarities of the meron pair correspond to a zero topological number ($N = 0$), our observation indicates that this meron pair creation process retains its initial $N = 0$ topology of the uniform in-plane magnetization.

In the region of $H \sim Hc$, the Py magnetization surrounding the disk region switches its direction abruptly by 180° through

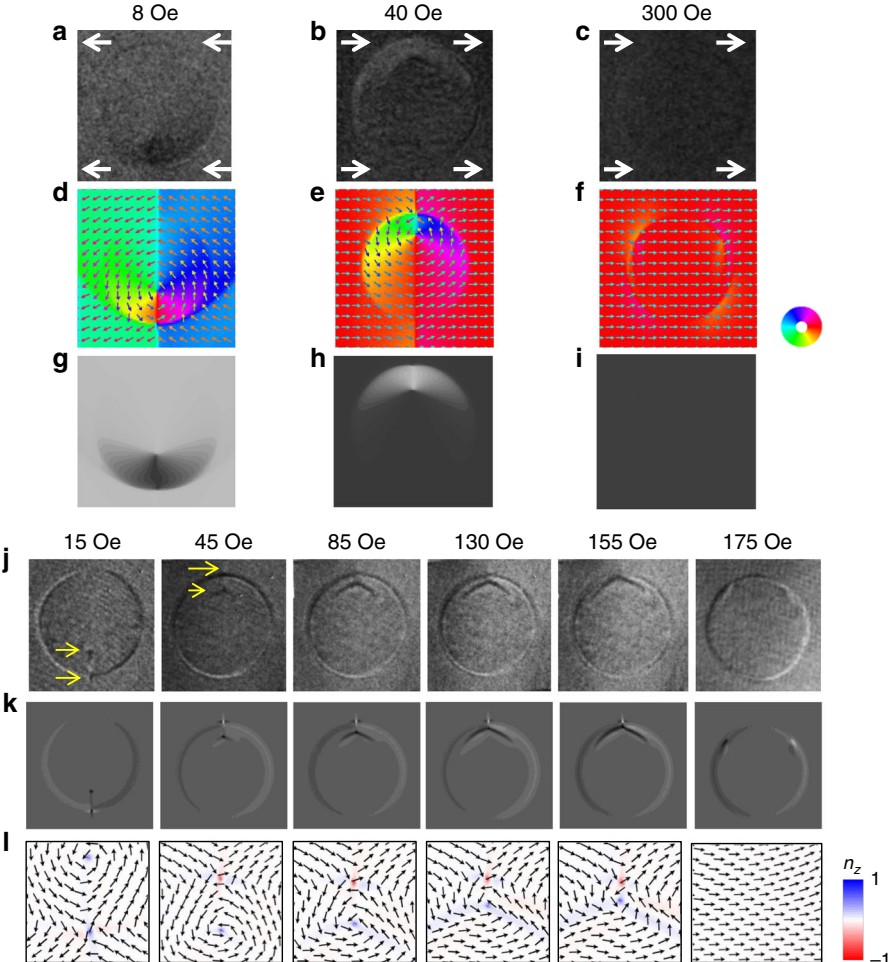

**Fig. 4 Magnetization process of a meron pair in the Py film. a–i** In-plane magnetic contrast at three different states of $H < +Hc$ (**a**, **d**, **g**), $+Hc < H < +Hs$ (**b**, **e**, **h**), and $H > +Hs$ (**c**, **f**, **i**) in the magnetization process. **a**, **b**, **c** The experimental results. The white arrows indicate the direction of surrounding Py magnetization. **d**, **e**, **f** Corresponding magnetization profiles obtained by simulation. **g**, **h**, **i** Corresponding thickness averaged $x$-component of the simulated magnetization profile in the Py film. **j**, **k**, **l** Out-of-plane magnetic contrast during the magnetization process. At the end of region (1) (e.g., the 15 Oe image in the figures), a meron pair is nucleated at the bottom edge of the disk area; after passing region (2) (e.g., the 45 Oe image in the figures), a new pair forms at the top edge; further increasing the magnetic field makes the distance between the two cores gradually reducing from ~300 nm toward zero (e.g., the 85–155 Oe images in the figures), and finally the meron pair is annihilated at the end of region (3) (e.g., the 175 Oe image in the figures). **j** The experimental images, where the yellow arrows in the first two images indicate the core position of vortices and antivortices; **k** thickness averaged $z$-component of the magnetization in the Py film in a simulation with the same field sequence; **l** zoomed in view of the magnetization profile close to the meron pairs at the bottom surface of Py from simulations, where the in-plane magnetization profile is plotted by arrows and the out-of-plane component of the magnetization is represented by the colors.

domain nucleation and propagation process. As mentioned earlier (Fig. 3), the resulting meron pair has equal probability of parallel and antiparallel core polarities. Therefore, we conclude that the topological number of the system is not preserved in this abrupt switching of the Py background magnetization.

In the region from $H > +Hc$ to $H = +Hs$, the Py magnetization surrounding the disk remains unchanged with the meron pair being annihilated by the magnetic field. Since we could have meron pairs with either parallel or antiparallel core polarities from the second region of $H \sim Hc$, we studied both cases ($|N| = 1$, $N = 0$) at $H = 25$ Oe, 85 Oe, 130 Oe, 155 Oe, 165 Oe, 175 Oe, and 200 Oe in sequence. Figure 5a shows the starting state ($H = 25$ Oe) of a bimeron ($N = -1$). With increasing the magnetic field, the vortex core moves toward the antivortex core and the bimeron is annihilated above 175 Oe. For a starting state ($H = 25$ Oe) of a meron–antimeron pair ($N = 0$; Fig. 5c), we

also observed that the vortex core moves toward the antivortex core with increasing field but the meron–antimeron pair is annihilated at 165 Oe, which is smaller than the 175 Oe for the bimeron case ($N = -1$). Considering that the final state has the topological number of $N = 0$, the higher annihilation field of bimeron ($N = -1$) in Fig. 5a than that of meron–antimeron pair ($N = 0$) in Fig. 5c suggests that topology[18,35,37,52] may have played a role in the meron pair annihilation process due to an additional topological barrier (Supplementary Note 5 and Supplementary Fig. 6). Since the small 10 Oe field difference could be easily overwhelmed by other effects, we further performed micromagnetic simulations and found that the topological effect in the meron annihilation process could be enhanced by changing the sample geometry (Supplementary Note 4 and Supplementary Fig. 5).

Our above result suggests that topological effect may have played a role in the magnetization process where the spin texture is deformed slowly and continuously (region 1 and 3), but not in

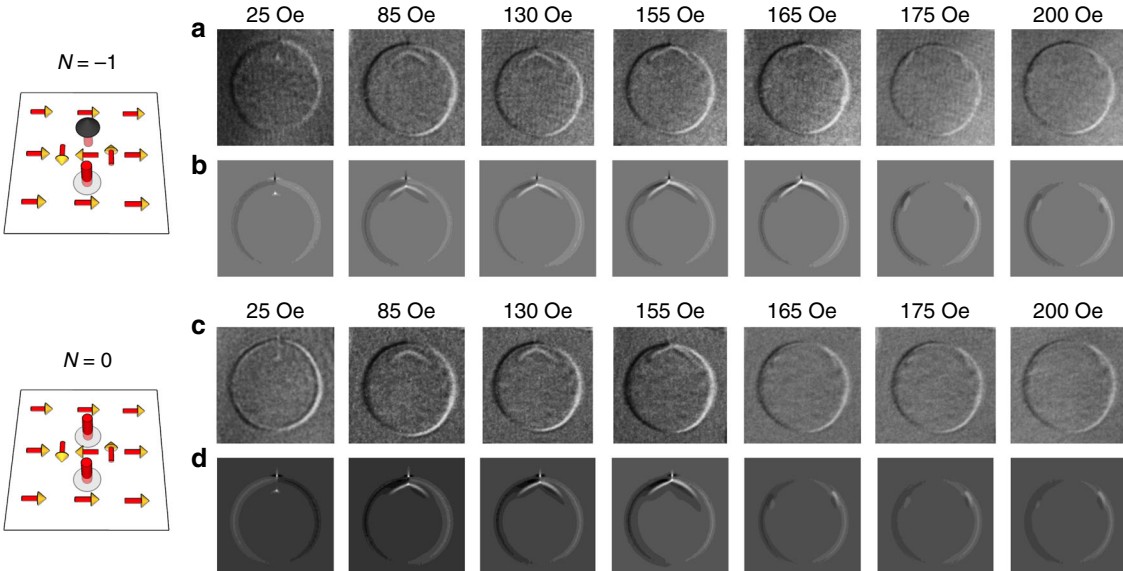

**Fig. 5 Annihilation of meron pairs of different topologies. a, b** Annihilation of a bimeron ($N = -1$), where the vortex and antivortex have opposite polarities. **c, d** Annihilation of a meron–antimeron pair ($N = 0$), where both the vortex and antivortex have the same polarity. **a, c** The experimental results; **b, d** thickness averaged $z$-component of the Py magnetization in a simulation with similar field sequence. The left schematic drawings show the starting topology of the two cases.

the magnetization switching process ($H - Hc$) where the global spin texture changes abruptly.

## Discussion

In summary, we develop a new method to create and stabilize a meron pair in a continuous Py film and determine the meron pair's topology directly during its creation and annihilation processes. Noticing that skyrmion has been the only nontrivial localized topological structure for $S^2$ spin space in 2D real space, the isolation of a single meron pair as a topologically equivalent skyrmion will open great opportunity for the study of topological effect in magnetic systems. Our method is quite general for in-plane magnetized systems with little restriction on the material choices (Supplementary Note 3 and Supplementary Fig. 4). The winding number invariance behind our methodology is in an analogy to the low-temperature phase of the Berezinskii–Kosterlitz–Thouless (BKT) transition[1], where vortices must pair with antivortices to maintain a global quasi long-range order. Different from the XY systems in the BKT transition, however, in magnetic thin films the existence of vortex/antivortex cores avoids the formation of topological defects, leading to a well-defined topology of the meron pair which is directly observed in this work. The universality of our method for the creation of meron pairs in in-plane magnetized materials removes the restriction of perpendicular magnetization and DMI, thus will extend topological spin structures to a much broader range of materials. In addition, for the same in-plane magnetization background, both $N = +1$ and $N = -1$ structures can be realized in our work (Fig. 3), as opposed to conventional skyrmions in non-centrosymmetric perpendicularly magnetized systems, whose $N$ is generally locked by the sign of DMI and background magnetization direction[5,12]. This is important to spintronic applications because usually two different stable states are required to encode one bit of data.

## Methods

**Sample fabrication**. An 80 nm thick Py film was grown on a 200 nm thick $Si_3N_4$ membrane using e-beam evaporation in an UHV chamber under a base pressure of $2.6 \times 10^{-10}$ Torr. After that, a shadow mask was loaded on top of the sample and a

layer of 40 nm Co was grown to form disk arrays with 1 μm disk radius and 3 μm center-to-center distance on top of the Py film. After removing the shadow mask, the sample was sputtered by $Ar^+$ ion for 10 min to remove any residual contaminations.

**Hysteresis loop measurement**. A longitudinal Magneto-Optic Kerr Effect (MOKE) was used to measure the hysteresis loops of the in-plane magnetization. The laser beam diameter is about 200 μm. Py hysteresis loop was obtained by aiming the MOKE laser beam outside the patterned area. Noticing the finite penetration depth of laser in metals (~10 nm), hysteresis loop of Co disk plus surrounding Py was obtained by aiming the MOKE laser beam at the patterned area.

**MTXM measurement**. The MTXM measurement were taken at the beamline 6.1.2 of the Advanced Light Source. For out-of-plane magnetization imaging, the sample was mounted to the normal incidence of the X-rays so that only the out-of-plane component of magnetization contributes to the magnetic contrast. For in-plane magnetization imaging, the sample was tilted with its normal 30º with respect to the X-ray direction so that the projection of the X-ray in the film plane picks up the corresponding magnetization. All the domain images were normalized by the corresponding saturation images to get enhanced magnetic contrast with reduced background noise.

**Micromagnetic simulations**. The open source micromagnetic code mumax3[53] was used for the simulations and the Object Oriented MicroMagnetic Framework (OOMMF)[54] was used for to plot the magnetization profiles. The material parameters were chosen as: $M_{Py} = 860$ emu cm$^{-3}$, $M_{Co} = 1400$ emu cm$^{-3}$, $A_{Py} = 1.3 \times 10^{-11}$ J m$^{-1}$, and $A_{Co} = 3.0 \times 10^{-11}$ J m$^{-1}$. The size of the simulation area is $3 \times 3$ μm and the mesh size is 11.7 nm × 11.7 nm × 3.75 nm. The use of these parameters was verified to give reasonable computation time while obtaining reliable results for the sample investigated in this work. An error threshold of $1.0 \times 10^{-9}$ (MaxErr = $10^{-9}$) was used as the convergence criteria for magnetization relaxation, which was actually adapted from the built-in Bogacki-Shampine solver of mumax3. The coupling constant between Co and Py was 1.05 erg cm$^{-2}$. Periodic boundary conditions were used to reduce the artifacts from edges of simulation areas. The magnetic field was applied at an angle of ~30 º with respect to the array direction to simulate the actual mounting direction of the sample (see Fig. 2a), and a 10 nm surface layer from both the Co surface and the Py surface (100 nm further than the disk area) was removed to better reflect the inevitable etching effect from $Ar^+$ ion sputtering.

## Data availability

The data that support the findings of this study are available from the corresponding author upon reasonable request.

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

## Acknowledgements

This work was primarily supported by US Department of Energy, Office of Science, Office of Basic Energy Sciences, Materials Sciences and Engineering Division under contract no. DE-AC02-05-CH11231 (van der Waals heterostructures program, KCWF16). Sample fabrication and hysteresis loop measurement were support by National Science Foundation grant no. DMR-1504568. C.H. and Z.Q.Q. acknowledge support from Future Materials Discovery Program through the National Research Foundation of Korea (no. 2015M3D1A1070467) and Science Research Center Program through the National Research Foundation of Korea (no. 2015R1A5A1009962). N.G. acknowledges support by the National Natural Science Foundation of China (no. 61505243). M.-Y.I acknowledges support by the National Research Foundation (NRF) of Korea (no. 2017R1A4A1015323). This work was also funded by DGIST R&D program of the Ministry of Science, ICT and future Planning (19-BT-02). J.W.C. acknowledges the KIST Institutional Program (2E29410), and the National Research Council of Science and Technology (NST) grant by MSIP (grant no. CAP-16-01-KIST). J.L. acknowledges financial support through National Key Research and Development Program of China (no. 2016YFA0300804 and 2017YFA0303303). This research used resources of the Advanced Light Source, which is a DOE Office of Science User Facility under contract no. DE-AC02-05CH11231.

## Author contributions

N.G., S.-G.J., M.-Y.I, J.W.C., J.L., and Z.Q.Q. designed and performed the experiments, analyzed the data, and wrote the paper. S.L. and H.-S.H. helped with the MTXM measurement. M.Y., Q.L, T.Y.W., K.-S.L., W.C., and C.H. were involved in the discussion of the results.

## Competing interests

The authors declare no competing interests.
