## [Peer Review File · Nature Communications]

Reviewers' comments:

Reviewer #1 (Remarks to the Author):

The authors discovered the topological meron pairs in a 2D in-plane magnetized system with deforming in-plane spin texture. In addition to directly image the meron pairs and hence confirm their topological features, the field-drive dynamics of meron pairs including the changes of topology and the annihilation process of meron pair have been demonstrated with combination of real-space observations and micromagnetic simulations. The scientific conclusion for the paper as well as the methodology for realizing the meron pair is very clear and impressive. The manuscript is rather well written, the figures are relatively clean, complete and correct as far as I can see. I recommend the publication of the present work in Nature communications after minor corrections as follows.

- Abstract

The meron is topologically equal to one-half of a skyrmion, however, their peripheral spin textures are importantly different: the spins of meron align in the in-plane while they are along the out-of-plane in the skyrmion. The authors should correct the relative phrase not only in the abstract part but also in the introduction.

- Experimental and results

Third paragraph

It is not clear for the creation of vortex pair with “---a flipping of the Py magnetization with a field of $H = + 35 \text{ Oe}$ ”. Please address this point to make more clearer.

- Imaging the magnetization process of the meron pairs

The end of first paragraph

“---observed recently in Py [44, 45].”

The reference 44 seems incorrect. Please check it carefully.

- Discussion part

Third paragraph

Line 9-line15

Authors claimed that the additional barrier may play a role to affect the field strength for the annihilation of bimeron. The physics view discussed here is very important. The additional simulations for the energy barrier for the topological transitions should be shown in the revised manuscript.

Reviewer #2 (Remarks to the Author):

In their paper, the authors describe a system of a Permalloy film with Co disks on to, which can stabilize in-plane magnetized meron pairs. The introductions clarifies the concept of merons and introduces the used sample system for the experimental work. This is supported by micro magnetic simulations. The experimental work is clear to an expert in the field, but I see a lack of clarity for a non expert in the field of x-ray microscopy. The Authors should considers this in the main text and give more details in the supplementary material.

The results are well presented and conclusions are sound.

Reviewer #3 (Remarks to the Author):

The authors use a variety of experimental techniques combined with micromagnetic modeling to create vortex-antivortex (or meron-antimeron) pairs in a Py thin film with in-plane magnetization. This is done by imprinting a vortex from a Co patterned on top of the Py film. The Co disk, being bounded,

can support a single vortex. The Py film is continuous and in order to conserve topological charge, the imprinted vortex (meron) comes with an antivortex (antimeron) in the film just outside of the region covered by the Co disk. The novelty of the work is the demonstration of creation of stable vortex-antivortex pairs (or meron-antimeron pairs) in a continuous thin film.

The experimental work is of high quality and well described. The micromagnetic modeling is straightforward modeling. However, the authors have omitted information that should be accompanying any description of micromagnetic modeling: the size of the domain used in the modeling, mesh size, convergence criteria, and the coupling between the Co and Py layers.

With this missing information, the work will be well described and self-contained. One can argue about whether or not the authors should include references to skyrmion lattices observed in films with inversion symmetry (so no DMI). However, my main hesitation in recommending the manuscript for publication in Nature Communication is the potential impact of the work. To be honest, I am on the fence about this. Yes, the work deserves to be published somewhere, but I am not sure Nature Communications is the "right" venue for this.

In order to answer clearly all comments of the reviewers, we write our reply (blue colored paragraphs) under each question of the reports in the following.

Reviewer #1:

The authors discovered the topological meron pairs in a 2D in-plane magnetized system with deforming in-plane spin texture. In addition to directly image the meron pairs and hence confirm their topological features, the field-drive dynamics of meron pairs including the changes of topology and the annihilation process of meron pair have been demonstrated with combination of real-space observations and micromagnetic simulations. The scientific conclusion for the paper as well as the methodology for realizing the meron pair is very clear and impressive. The manuscript is rather well written, the figures are relatively clean, complete and correct as far as I can see. I recommend the publication of the present work in Nature communications after minor corrections as follows.

We thank the referee for his/her recommendation to publish our paper.

- Abstract

The meron is topologically equal to one-half of a skyrmion, however, their peripheral spin textures are importantly different: the spins of meron align in the in-plane while they are along the out-of-plane in the skyrmion. The authors should correct the relative phrase not only in the abstract part but also in the introduction.

We agree with the reviewer that a meron and one-half skyrmion could have very different peripheral spin textures even though they are topologically equivalent to each other. Following the reviewer's suggestion, we revised in the abstract and introduction by referring meron as a topologically equivalent spin texture to one-half skyrmion.

- Experimental and results

Third paragraph

It is not clear for the creation of vortex pair with “---a flipping of the Py magnetization with a field of $H = +35$ Oe”. Please address this point to make more clearer.

In the revised manuscript, we described the “flipping” more clearly as switching the surrounding Py magnetization from -x direction to +x direction by changing the magnetic field from $H=-430$ Oe to $H=+35$ Oe, where the $H=-430$ Oe saturates both the Py disk area and the surrounding area magnetizations to the -x direction and the $H=+35$ Oe is greater than the Py coercivity to switch the surrounding Py magnetization from -x to +x direction but less than the saturation field of the vortex. This process creates meron pairs with topological number of $N=0$, and ± 1 . As shown later on the topological effect in the main text, another way of creating a meron pair is to reduce the field from negative saturation to $H=-15$ Oe where the -15 Oe is weak enough for Py to form a vortex at the disk region but remaining Py magnetization in the original -x direction outside the disk region (e.g., Fig. 4b of the main text). This process does not change the surrounding Py magnetization direction, and as shown in the paper preserves the topology to produce $N=0$ meron pair only.

- Imaging the magnetization process of the meron pairs

The end of first paragraph

“---observed recently in Py [44, 45].”

The reference 44 seems incorrect. Please check it carefully.

We thank the referee for finding this error, and have deleted Ref 44 in the revised manuscript.

- Discussion part

Third paragraph

Line 9-line15

Authors claimed that the additional barrier may play a role to affect the field strength for the annihilation of bimeron. The physics view discussed here is very important. The additional simulations for the energy barrier for the topological transitions should be shown in the revised manuscript.

We thank the referee for this insightful comment, and agree that the physics here deserves more discussion.

In fact, transition between two topological states is an active and difficult research topic. In particular, a transition from an initial topological magnetic state ($N=1$ or $N=0$) to a ferromagnetic (FM) state (quasi-single-domain state, $N=0$) usually needs to overcome an energy barrier which separates the two local energy minimum (or called valley in energy landscape). As shown by many recent theoretical works [*Phys. Rev. B* **65**, 024414 (2001), *Nat. Commun.* **6**, 8455 (2015), *Phys. Rev. B* **93**, 214412 (2016), *Sci. Rep.* **7**, 4060 (2017)], the difficulty on this subject is that there exist infinite paths (e.g., magnetization profile sequences) to connect these two local states, making it impossible to find the exact minimal energy path linking the skyrmion state to the FM state. Different methods have been developed aiming to approach the minimal energy path in order to estimate the energy barrier [*J. Chem. Phys.* **113**, 9978-9985 (2000), *IEEE Trans. Magn.* **54**, 7206105 (2018), *Phys. Rev. Lett.* **121**, 197202 (2018)]. One practical approach is to take the series of magnetization profiles from the actual annihilation process as the transition path and calculate their energies at fixed system parameters to obtain the corresponding energy barrier. For our case, this corresponds to the calculation of the energy densities of the whole sequence of magnetization profiles for the annihilation processes of the bimeron state ($N=-1$) and the meron-antimeron pair states ($N=0$) shown in Supplementary Figure 5, but with the Zeeman energy calculated using a prefixed magnetic field (because this field is a system parameter upon which the energy landscape depends on), and obtain the corresponding energy barrier which separates the bimeron state (or the meron-antimeron pair) and the quasi-single-domain state (FM state).

Since our goal is to compare the difference between the bimeron and the meron-antimeron pair, we calculated the energy densities of the series of the magnetization profiles shown in Supplementary Figure 5 (plus many detailed intermediate steps) with the Zeeman energy calculated at the field of $H=600$ Oe which is just below the annihilation field of the meron pairs to best present the energy barrier that separates the two energy valleys in the configuration space. Fig. R1 depicts the calculated energy density for bimeron (E_1 , red color) and meron-antimeron pair (E_0 , blue) along the evolution path of the magnetization profiles (Supplementary Figure 5). Each evolution shows that the initial state needs to overcome an energy barrier ($\Delta E_0 = 1.18 \text{ kJ/m}^3$ for meron-antimeron pair, and $\Delta E_1 = 1.53 \text{ kJ/m}^3$ for

bimeron) to collapse into the final FM state (quasi-single domain state). The role played by topology can be seen by comparing the energy barriers between these two paths, which shows that the bimeron path has a higher energy barrier than the meron-antimeron pair path by $\Delta E_1 - \Delta E_0 = 0.35 \text{ kJ/m}^3$. One can interpret this additional energy barrier as the manifest of topological effect since it will take a higher magnetic field to trigger the topological transition (N=-1 to N=0) than the non-topological transition (N=0 to N=0). We note that such manifestation of topology is in accordance with the result of previous works [*Phys. Rev. Lett.* **97**, 177202 (2006), *Nat. Commun.* **5**, 4704 (2014) and *Phys. Rev. Lett.* **118**, 267203 (2017)] which also show the different annihilations of N=1 and N=0 states into the final FM state. We added the above simulation result and discussion as the revised Supplementary Note 5.

Fig. R1 The energy densities of the bimeron (N=-1) and the meron-antimeron pair (N=0) along the evolution paths to the final FM state (quasi single domain) with the Zeeman energy calculated at H=600 Oe. The higher barrier for the bimeron (ΔE_1) than the meron-antimeron pair (ΔE_0) is a manifest of topological effect in the transition. The horizontal axis labels the magnetization profiles at the corresponding fields in Supplementary Figure 5 along the evolution path (e.g., the 500 magnetization profile corresponds to the magnetization profiles at H=500 Oe in Supplementary Figure 5). In addition, the right column gives representative zoomed in Py magnetization profiles at the three labeled locations (A,B,C) in the transition path.

Reviewer #2:

In their paper, the authors describe a system of a Permalloy film with Co disks on to, which can stabilize in-plane magnetized meron pairs. The introductions clarifies the concept of merons and introduces the used sample system for the experimental work. This is supported by micro magnetic simulations. The experimental work is clear to an expert in the field, but I see a lack of clarity for a non expert in the field of x-ray microscopy. The Authors should considers this in the main text and give more details in the supplementary material.

The results are well presented and conclusions are sound.

We thank the reviewer for suggesting a detailed description on the x-ray microscope in order for none experts to better understand the result of the paper. To address this suggestion, we added a brief description to non-experts on the principle of MTXM in the main text, and a detailed discussion in Supplementary Information (Section 1). The following shows where we added in the paper.

In the main text, we added the following paragraph before presenting Fig. 2c and 3d:

To experimentally confirm the methodology of the vortex imprinting, one needs to measure the magnetization profiles of the Co disk and the Py film separately. For this purpose, we used full-field

Magnetic Transmission soft X-ray Microscopy (MTXM), which enables element-resolved magnetic imaging (e.g., imaging Co and Py separately) by tuning the x-ray photon energy to the 2p core level absorption energy of the corresponding element [Mater. Today 9, 26-33 (2006)]. The MTXM spatial resolution could reach 25nm by the state-of-the-art X-ray optics of the so-called Fresnel zone plates [Opt. Express 17, 17669-17677 (2009)] (Supplementary Figure 1). The MTXM magnetic contrast comes from the X-ray magnetic circular dichroism (XMCD) mechanism [Coord. Chem. Rev. 277, 95-129 (2014)]: the absorption of a circularly polarized x-ray by a ferromagnetic sample depends on the angle between the magnetization direction of the sample and the photon spin direction of the x-ray. At a fixed incident angle of a circularly polarized x-ray beam, different magnetization directions of a magnetic domain then give rise to different contrasts (dark/grey/bright) of the MTXM image.

In addition, when the magnetic images were discussed for the first time (i.e., Fig. 2c and d for in-plane images, and Fig. 3 for out-of-plane images), we added explanation on the corresponding relation between the dark/grey/bright contrasts in the images and the magnetization directions of the sample. In this way, readers can easily understand the magnetic images even without the knowledge on the x-ray microscope.

In the Supplementary Information, the setup of the full-field MTXM is described by a schematic drawing in Figure R2. The x-ray coming from the synchrotron source is focused by a zone plate (an x-ray version of optical lens) onto the sample placed closely after a pinhole for illumination. After transmitting through the sample, the x-ray is focused again by another zone plate onto the CCD camera. The setup is essentially an x-ray version of conventional optical microscope, with the visible light source replaced by the synchrotron x-rays, the optical condenser lens replaced by the x-ray condenser zone plate, and the optical objective lens replaced by the x-ray micro zone plate accordingly. Together with the X-ray Magnetic Circular Dichroism (XMCD) effect (see below), the MTXM achieves element-resolved magnetic imaging which has been widely used for the imaging of magnetic domains in magnetic nanostructures.

Figure R2, Schematic drawing of MTXM setup. This is essentially an x-ray version of a conventional optical microscope.

The XMCD effect is equivalent to the Magneto-Optic effect except the XMCD measurement is performed using x-rays with the x-ray photon energy equal to the L edge absorption energy of 3d transition metals so that the absorption is element specific. Akin to the Magneto-Optic Effect, left- and right circular polarized x-rays will be absorbed differently by a ferromagnetic material (FM) whose magnetization is parallel to the x-ray propagation direction. Equivalently, for a fixed circularly polarized x-rays, the ferromagnetic material will absorb the x-rays differently, depending on its magnetization parallel or antiparallel to the x-ray propagation direction (or precisely speaking, depending on the projection the sample magnetization to the x-ray photon spin direction). For general case, the projection of the FM magnetization to the photon spin of a circularly polarized x-rays determines the XMCD signal so that magnetic domains with magnetization at different angles with respect to the x-ray photon spin would exhibit different contrast

due to the XMCD effect.

As shown in Figure R3a, under the experimental condition of 30° sample tilting (i.e., 60° x-ray incident angle), the domains with magnetization pointing toward $+x$ and $-x$ directions will have dark and bright contrast, respectively, with magnetization pointing to $\pm y$ direction having grey contrast. Then a vortex state in an FM disk will exhibit a gradually changing contrast due to its curling magnetization (Supplementary Figure R3b). For normal incidence of x-rays, out-of-plane magnetizations in $+z$ and $-z$ directions show dark and bright contrast, while all the in-plane magnetizations give a grey contrast because they are all perpendicular to the x-ray photon spin (Figure R3c). Therefore as schematically shown in Figure R3d, normal incidence of x-rays is best to image the vortex core polarities.

Figure R3, Magnetic contrast in MTXM. **a**, By mounting the sample with a 30° sample tilting, different in-plane magnetization directions in a magnetic domain absorb a circular polarized x-ray beam differently due to the XMCD effect, leading to different magnetic contrasts (dark/bright/grey). The 1st row illustrates the incident direction of x-ray, and the 2nd and 3rd rows show different contrasts (dark/bright/grey) from different magnetization directions (represented by the yellow arrows). **b**, An example of in-plane magnetic contrast of a vortex state in an FM disk. The left figure illustrates the in-plane magnetization profile, and the right illustrates the corresponding contrast in the MTXM image. Note here only 4 directions ($+x$, $-x$, $+y$, $-y$) of magnetization and 3 magnetic contrasts (dark, grey, bright) are shown for simplicity. In real case, however, both the magnetization and the corresponding contrast vary continuously as shown in Fig. 2 of the main text. **c**, By mounting the sample normal to incident x-ray, only the out-of-plane magnetization will give bright or dark contrast, and all the in-plane domains will show grey contrast. The 1st row illustrates the incident direction of x-ray, and the 2nd and 3rd show the different contrast (represented by colors in the plane) of domains with different magnetization directions (represented by the yellow and red arrows for the in-plane and out-of-plane magnetizations, respectively). **d**, An example of out-of-plane magnetic contrast of a vortex state in an FM disk. The left figure illustrates the magnetization profile, and the right illustrates the corresponding contrast in the MTXM image.

Reviewer #3:

The authors use a variety of experimental techniques combined with micromagnetic modeling to create vortex-antivortex (or meron-antimeron) pairs in a Py thin film with in-plane magnetization. This is done by imprinting a vortex from a Co patterned on top of the Py film. The Co disk, being bounded, can support a single vortex. The Py film is continuous and in order to conserve topological charge, the imprinted vortex (meron) comes with an antivortex (antimeron) in the film just outside of the region covered by the Co disk. The novelty of the work is the demonstration of creation of stable vortex-antivortex pairs (or meron-antimeron pairs) in a continuous thin film.

The experimental work is of high quality and well described. The micromagnetic modeling as straightforward modeling. However, the authors have omitted information that should be accompanying any description of micromagnetic modeling: the size of the domain used in the modeling, mesh size, convergence criteria, and the coupling between the Co and Py layers. With this missing information, the work will be well described and self-contained. One can argue about whether or not the authors should include references to skyrmion lattices observed in films with inversion symmetry (so no DMI).

Following the reviewer's suggestion, we add the missed information in the Methods part of the revised manuscript. Specifically, the size of the simulation area is $3\mu\text{m}\times 3\mu\text{m}$ and the mesh size is $11.7\text{nm}\times 11.7\text{nm}\times 3.75\text{nm}$. The use of these parameters was verified to give reasonable computation time while obtaining reliable results for the sample investigated in this work. An error threshold of 1.0×10^{-9} (MaxErr= 10^{-9}) was used as the convergence criteria for magnetization relaxation which was actually adapted from the built-in Bogacki-Shampine solver of Mumax³. The coupling constant between Co and Py was 1.05 erg/cm^2 .

Regarding to "whether or not the authors should include references to skyrmion lattices observed in films with inversion symmetry (so no DMI)", we added new references in the revised manuscript to make the topic more complete. Specifically, [*Nat. Phys.* **7**, 713-718 (2011)] reports skyrmion lattice in monolayer Fe film on Ir(111) surface. Although DMI might exist in this system, the four-spin interaction also plays an important role, raising the possibility of skyrmion lattices without DMI. [*Nat. Commun.* **6**, 462 (2015)] reports artificial skyrmion lattices without DMI. [*Proc. Natl. Acad. Sci.* **109**, 8856-8860 (2012)] and [*Phys. Rev. B.* **95**, 024415 (2017)] report skyrmion lattices due to dipole interactions. Skyrmion lattices resulting from frustrating exchange interactions were numerically observed in [*Phys. Rev. Lett.* **108**, 017206 (2012)] and those with geometric frustrations were experimentally observed in [*Science* **365**, 914-918(2019)]. [*Nat. Phys.* **7**, 713-718 (2011)] and [*Nat. Commun.* **6**, 462 (2015)] were already cited in our original manuscript, and we've added the other 4 references in the revised manuscript as Ref. 43-46.

However, my main hesitation in recommending the manuscript for publication in Nature Communication is the potential impact of the work. To be honest, I am on the fence about this. Yes, the work deserves to be published somewhere, but I am not sure Nature Communications is the "right" venue for this.

We respect the reviewer's opinion. After reading our manuscript again, we realize that we perhaps didn't emphasize and explain clearly the importance and potential impact of this work. Therefore we added more discussion in the "Summary" section to further emphasize the following points. We hope that this added discussion has addressed the reviewer's concern on the potential impact of our work.

- (1) From the point of view of fundamental research, it has been one of the major and most active research subjects in today's condensed matter physics to search and explore topological objects

in materials. For S^2 spin space in 2D real space, skyrmion has been the only nontrivial localized topological structure. That is largely why skyrmion research has been one of the most active and important topic in magnetic research in the last decade.

- (2) There has been a great effort in searching for different topological spin textures even through these different spin textures are topologically equivalent (e.g., different spin textures with the same skyrmion number). For example, it is now recognized that there exist Bloch-type and Néel-type skyrmions in *perpendicular* magnetized systems. In this work, we demonstrate that a new type of spin texture topologically equivalent to a non-trivial skyrmion ($N=\pm 1$) could exist in *in-plane* magnetized system as a meron pair, which has not been observed unambiguously before [*Nature* **564**, 43-44 (2018)]. To our knowledge, our work is the first isolation of a single meron pair in a continuous film which was stabilized locally, investigated directly, with their topologies determined unambiguously.
- (3) Conventional magnetic skyrmions exist only in a small group of materials mainly because of the restriction in perpendicularly magnetized thin films. This limitation prohibits further exploration of skyrmions, particularly recognizing that magnetic thin films have an inherent tendency to be in-plane magnetized because of the inevitable demagnetization field. Our construction of meron pair as a topologically equivalent skyrmion in in-plane magnetized thin films removes the requirement of perpendicular magnetization, thus extends this topological structure to a much broader range of materials.
- (4) Finally, from the practical point of view, our work shows that for the same magnetic background (e.g., magnetization in +x direction), both $N=+1$ and $N=-1$ meron pairs can be realized (Fig. 3 of the main text). This is important to spintronics application because it usually requires two different stable states to encode one bit of data. In contrast, the sign of skyrmion number of conventional skyrmion in perpendicularly magnetized systems is locked by the background magnetization direction (e.g., +z direction) [*Nat. Nanotechnol.* **8**, 899–911 (2013), *Nature* **548**, 561-566 (2017)], which was already reported to cause problems in encoding information [*IEEE Electron Device Lett.* **37**, 924-927 (2016)]. Although it was anticipated that the recent observation of skyrmions in centrosymmetric systems might alleviate this problem by hosting both skyrmions and antiskyrmions simultaneously [*Science* **365**, 914-918(2019)], it is unclear on when a clear experimental observation and demonstration would be achieved in the future.

REVIEWERS' COMMENTS:

Reviewer #1 (Remarks to the Author):

The authors have provided thoughtful responses to all of the issues that I raised in my first review, and I believe the new clarifications added to this revision make the work suitable for publication in Nature Communications.

Reviewer #1 (Remarks to the Author):

The authors have provided thoughtful responses to all of the issues that I raised in my first review, and I believe the new clarifications added to this revision make the work suitable for publication in Nature Communications.

We thank the referee for his/her recommendation to publish our paper.